# The Detection and Negative Reversion of Circulating Tumor Cells as Prognostic Biomarkers for Metastatic Castration-Resistant Prostate Cancer with Bone Metastases Treated by Enzalutamide

**DOI:** 10.3390/cancers16040772

**Published:** 2024-02-13

**Authors:** So Nakamura, Masayoshi Nagata, Naoya Nagaya, Takeshi Ashizawa, Hisashi Hirano, Yan Lu, Hisamitsu Ide, Shigeo Horie

**Affiliations:** 1Department of Urology, Juntendo University Graduate School of Medicine, Tokyo 1138-431, Japan; so-nakamura@juntendo.ac.jp (S.N.); m-nagata@juntendo.ac.jp (M.N.); nanagaya@juntendo.ac.jp (N.N.); ashizawa@juntendo.ac.jp (T.A.); hhirano@juntendo.ac.jp (H.H.); lyan@juntendo.ac.jp (Y.L.); 2Department of Advanced Informatics of Genetic Diseases, Digital Therapeutics, Juntendo University Graduate School of Medicine, Tokyo 1138-421, Japan; h.ide.me@juntendo.ac.jp

**Keywords:** circulating tumor cells (CTCs), enzalutamide, bone metastases, castration-resistant prostate cancer, androgen receptor signaling inhibitor

## Abstract

**Simple Summary:**

The development of biomarkers that can predict the effectiveness of treatments for metastatic castration-resistant prostate cancer (mCRPC) has been a recent challenge. We focused on the efficacy of circulating tumor cell (CTC) status as a prognostic biomarker after enzalutamide administration. A retrospective subgroup analysis and prognostic survey were conducted on 43 patients with mCRPC and bone metastases. The results showed that patients with no detected CTCs at baseline showed significantly longer overall survival (OS) than those with CTCs at baseline. Furthermore, patients who exhibited a negative reversion of CTC status during enzalutamide treatment had significantly longer OS compared to patients with persistent CTC positivity. Achieving CTC-negative reversion during treatment for mCRPC with bone metastases was associated with improved long-term OS. Chronological measurement of CTC status might be clinically useful in the treatment of mCRPC.

**Abstract:**

Enzalutamide is a second-generation androgen receptor inhibitor that increases overall survival (OS) rates in patients with metastatic castration-resistant prostate cancer (mCRPC). This study evaluates the efficacy of circulating tumor cell (CTC) status as a prognostic biomarker following enzalutamide administration. A retrospective subgroup analysis and prognostic survey were conducted on 43 patients with mCRPC and bone metastases treated in Juntendo University-affiliated hospitals from 2015 to 2022. Patients were treated with 160 mg enzalutamide daily. CTC analyses on blood samples were performed regularly before and every three months after treatment. The relationship between the patients’ clinical factors and the OS rate was analyzed using the log-rank test; the median OS was 37 months. Patients with no detected CTCs at baseline showed significantly longer OS than those with detectable CTCs at baseline. Furthermore, patients demonstrating negative reversion of CTCs during enzalutamide treatment had significantly longer OS than patients with CTC-positivity. Two biomarkers—higher hemoglobin at baseline and achieving negative reversion of CTCs—were significantly associated with prolonged OS. This study suggests that patients achieving CTC-negative reversion during treatment for mCRPC with bone metastases exhibit improved long-term OS. Chronological measurement of CTC status might be clinically useful in the treatment of mCRPC.

## 1. Introduction

Enzalutamide (ENZ) targets several androgen receptor signaling pathway steps, which are the key drivers of prostate cancer growth. It inhibits androgen signaling by binding to receptors, inhibiting nuclear translocation, binding to androgen response elements, and recruiting coactivators [1]. ENZ prolongs progression-free survival (PFS) and overall survival (OS) rates in patients with metastatic castration-resistant prostate cancer (mCRPC), regardless of previous administration of chemotherapy [2,3]. It has been available in Japan for the treatment of mCRPC since 2014.

Biomarkers for predicting the prognosis and efficacy of ENZ treatment for mCRPC have not been widely established. Several prognostic models of mCRPC have been developed to estimate long-term survival using common clinical indicators, such as prostate-specific antigen (PSA) levels [4,5,6].

With advances in the treatment of advanced prostate cancer, there have been important developments in the field of liquid biopsy beyond the tumor marker PSA, which may help manage the heterogeneous disease of prostate cancer in recent years [7]. Antonarakis et al. reported that the detection of circulating tumor cells (CTCs) using liquid biopsy is a promising biomarker [8]. Liquid biopsies allow the carrying out of analyses using non-solid biological tissues and are thereby less invasive, painless, and inexpensive compared to tissue biopsies and can be carried out repeatedly [9]. CTC-negative patients at the time of treatment administration demonstrate superior clinical outcomes to those of CTC-positive patients [8,10]. Androgen receptor splice variant 7 (AR-V7) in CTCs of patients with mCRPC is a potential biomarker to predict the development of drug resistance to ENZ and abiraterone [8,10,11]. Unlike antiandrogen treatments, taxanes are effective for mCRPC, irrespective of AR-V7 status [12,13], which has been confirmed in the Asian population as well [14]. Therefore, to optimize treatment selection and avoid severe adverse effects, genetic analysis of CTCs can be used for personalized treatment for mCRPC.

Since CTCs can be collected from a patient’s peripheral blood, CTC analyses are minimally invasive and can be performed relatively easily in any laboratory. Therefore, we conducted this study to evaluate the usefulness of CTC analysis in the clinical setting of mCRPC. In a previous study conducted at our facilities and affiliated hospitals, we analyzed PFS in mCRPC patients with bone metastases treated with ENZ and demonstrated that a negative CTC status at baseline before ENZ treatment was a significant predictor for ENZ efficacy [10]. It has already been reported that chronological monitoring of CTCs is very effective for evaluating possible treatments and their efficacy in mCRPC [15,16,17]. The purpose of the current study is to establish the relationship between the chronological detection of CTCs during ENZ treatment and OS by setting a longer observation period and performing CTC detection at 3 months after ENZ introduction.

## 2. Materials and Methods

### 2.1. Patients and Study Design

Sixty mCRPC patients with bone metastases treated at Juntendo University Hospital between 2015 and 2022 were included in this study. All patients were administered 160 mg ENZ daily. Blood samples and CTC analyses were performed regularly before and every three months after treatment at Juntendo University Hospital. Patients were grouped based on their CTC status prior to ENZ administration. As shown in the flowchart in Figure 1, among patients who were CTC-positive at baseline, we compared patients who showed negative reversion of CTCs during ENZ treatment with those who remained CTC-positive despite treatment. Negative reversion was defined as a phenomenon in which CTCs dropped to undetectable levels during the course of treatment, although they were detected at pretreatment baseline. The primary endpoint was OS, a measure of the time from the date of ENZ administration to the date of death or loss of follow-up. Statistically significant clinical factors were identified in these two groups using univariate analysis. Specifically, patients’ clinical background factors, such as age, Gleason Score (GS), previous treatment, time-to-CRPC, laboratory data, changes in bone scan index (BSI), and the presence of CTCs, were analyzed.

### 2.2. CTC Analyses

We used the AdnaTest (QIAGEN, Hilden, Germany) to detect CTCs according to the manufacturer’s protocol [8,11,17]. The patient’s blood (5 mL) was drawn into EDTA-3K collection tubes, followed by RNA extraction with antibody-conjugated magnetic beads using AdnaTest ProstateCancerSelect. mRNA was extracted using the AdnaTest ProstateCancerDetect kit. The extracted mRNA was subjected to reverse transcription using the Sensiscript Reverse Transcriptase Kit (QIAGEN). The expression of PSA, AR-V7, and AR in CTCs was examined by reverse transcription polymerase chain reaction (RT-PCR). The AdnaTest PrimerMix ProstateDetect was used for the amplification of PSA (PCR conditions: 95 °C for 15 min, 42 cycles of 94 °C for 30 s, 61 °C for 30 s, 72 °C for 30 s, followed by 10 min of extension). The AdnaTest PrimerMix AR-Detect was used to amplify AR (PCR conditions: 95 °C for 15 min, 35 cycles of 94 °C for 30 s, 60 °C for 30 s, 72 °C for 60 s, followed by 10 min of extension).

Our experiments confirmed that the samples tested positive for PSA. Thus, we concluded that PSA positivity is a common denominator and defined successful CTC detection as positive PSA expression. The primer set and PCR conditions for AR-V7 RT-PCR are as follows: AR-V7 primer set designed to yield 125-bp AR-V7-specific band: 5′-CCATCTTGTCGTCTTCGGAAATGTTA-3′ and 5′-TTTGAATGAGGCAAGTCAGCCTTTCT-3′; PCR conditions: 95 °C for 5 min, 39 cycles of 95 °C for 10 s, 58 °C for 30 s, 72 °C for 30 s, followed by 10 min of extension. The amplified PCR products were electrophoresed and visualized using the DNA 1 K Experion Automated Electrophoresis System (Bio-Rad, Hercules, CA, USA). To evaluate gene expression, the fluorescence intensity scale was set to “scale to local” (default setting), and under this condition, any visible bands with detectable peaks were considered positive.

### 2.3. Statistical Analyses

Statistical analyses were performed using Fisher’s exact test for categorical variables. The student’s *t*-test was used to analyze normally distributed continuous variables. The Mann–Whitney U test was used for non-normally distributed continuous variables. OS analyses were performed using Kaplan–Meier plots, and differences were compared using the log-rank test. Univariate analyses were performed using log-rank tests. The cut-off value for each factor was the median value. Multivariate analysis used a bivariate COX proportional hazards model. Statistical significance was defined as *p* < 0.05. All statistical analyses were performed with the EZR software (R version 3.4.1 (2017-06-30)) for medical statistics, which is based on R and a modified version of R commander designed to add statistical function and frequently used in biostatistics [18].

### 2.4. Ethics Statement

This study was approved by the Institutional Review Board of Juntendo Hospital (approval numbers: 14-052 and 15-060), and all experiments were performed in accordance with approved guidelines. All participants provided written informed consent.

## 3. Results

A total of 43 of the 60 mCRPC patients were treated with ENZ without dose reduction or discontinuation of ENZ and underwent serial CTC analyses before and during treatment with ENZ. The baseline characteristics of patients before ENZ administration are shown in Table 1. The median age at diagnosis of prostate cancer was 73.0 (interquartile range (IQR): 69.0–78.0), and the median initial PSA value was 119.2 (IQR: 36.3–658.4) ng/mL. The Gleason score of the biopsy when they were diagnosed with prostate cancer was 8 or more in 77% of cases (33/43). The median time-to-CRPC, time from diagnosis of prostate cancer to CRPC, was 16.0 (IQR: 10.0–39.0) months, and the median PSA value at baseline before ENZ administration was 9.6 (IQR: 4.1–36.6) ng/mL. ENZ was used as the first-line treatment for mCRPC in 84% (36/43) of cases.

The baseline characteristics of the groups classified based on the absence or presence of CTCs before ENZ administration are shown in Table 1. Two factors, PSA level and BSI upon bone scintigraphy before ENZ administration, were significantly higher in the CTC-positive group (*p* = 0.004 and *p* = 0.008, respectively).

The observation period was from ENZ administration until death or the end of the study period, with an overall average observation period of 37 months (range 5–73 months). During the final observation period of this study, stratified analysis based on baseline CTC positivity revealed a median observation period of 31 months (range: 5–72) for the CTC-positive group and 44 months (range: 21–73) for the CTC-negative group. In the Kaplan–Meier curve analyzed by the log-rank test, the baseline CTC-negative group had significantly longer OS than the baseline CTC-positive group (median OS: not reached (NR) (44 months estimated) vs. 31 months; (HR, 3.03 [95% CI; 1.11–8.23]; *p* = 0.023; Figure 2). The mortality events for overall survival rates stratified by baseline CTC negativity/positivity were as follows: In the entire cohort, there were 22 deaths out of 43 cases (51%). Within the CTC-positive group, the mortality events were 17 out of 26 cases (65%), while in the baseline CTC-negative group, the events were 5 out of 17 cases (29%). The number of CTC-positive patients at baseline was 60.5% (26/43) of cases (Figure 1, Table 1), while 46.2% of patients showed “negative reversion” of CTCs during treatment with ENZ (12/26). In addition, 53.8% (14/26) of CTC-positive patients remained as such during ENZ treatment. Only two cases converted from negative to positive despite ENZ treatment. Regarding the presence or absence of AR-V7 at baseline, AR-V7 was positive in 4/14 cases in the continuous CTC positive group and in 0/12 cases in the CTC negative conversion group.

Table 2 shows the differences in each background factor between the group that showed negative reversion of CTCs during ENZ treatment and the group that continued positive CTC detection during ENZ treatment. Bone Scan Index (BSI) in bone scintigraphy at baseline before ENZ administration was significantly lower in the patients achieving negative reversion of CTCs during ENZ treatment (*p* = 0.021). Hemoglobin at baseline was significantly higher in the patients achieving negative reversion of CTCs (*p* = 0.047). All cases that achieved negative reversion of CTCs during ENZ treatment were negative for lymph node metastasis at baseline (0/12), which was a statistically significant difference compared to those that remained positive for CTC detection (*p* = 0.017).

In the Kaplan–Meier curve of patients with baseline CTC-positive analyzed by the log-rank test, the group that achieved negative reversion of CTCs during treatment with ENZ had significantly prolonged OS compared to the group that remained CTC-positive (median OS: not reached (47 months estimated) vs. 24 months; HR: 3.97 [95% CI; 1.36–11.67]; *p* = 0.007; Figure 3a). In this study, although the group whose CTC was negative at baseline and remained negative during the observation period also did not reach the median OS during the observation period (Figure 3b), this group and the patients who achieved negative reversion from positive CTC to negative found no statistically significant difference in OS. In contrast, only two cases showed positive conversion of CTC status from baseline negative to positive during ENZ treatment. One patient was alive during the observation period, and one patient died of cancer (27 months later).

Each factor that significantly contributed to OS prolongation was identified using univariate analysis in the CTC-positive group at baseline (Table 3). According to the log-rank test, two statistically significant clinical prognostic factors were identified: higher hemoglobin levels at baseline (*p* = 0.026) and negative reversion status of CTCs during ENZ treatment (*p* = 0.007). Patients who achieved a −30% and −50% PSA reduction after 3 months of treatment tended to have a longer OS, but the difference was not statistically significant (*p* = 0.353 and 0226, respectively). Bivariate COX proportional hazards analysis was performed for high hemoglobin at baseline and CTC-negative reversion, which were significant factors in Table 3. The results showed that negative CTC reversion was a stronger factor in prolonging OS than high hemoglobin levels (*p* = 0.029 vs. 0.060) (Table 4).

## 4. Discussion

In our previous study, we demonstrated the efficacy of ENZ in treating mCRPC with bone metastases and showed that the presence or absence of CTCs at baseline correlated with the prolongation of PFS by ENZ treatment [10]. In this study, by analyzing a subgroup from the previous study, we first demonstrated that the absence of CTC detection at baseline might contribute to prolonged OS (Figure 2) and showed the efficacy of chronological analyses of CTC in 43 patients with mCRPC. As a result, we showed that negative reversion in CTC is a strong prognostic biomarker for longer OS (Figure 3a, Table 3 and Table 4).

Our analysis showed that CTC detection at baseline was significantly correlated with a high BSI upon bone scintigraphy and high PSA levels (Table 1). All patients had bone metastasis, and since bone metastasis is generally considered a hematogenous metastasis, in cases with a large amount of bone metastasis, we expected to detect more prostate cancer cells in the blood. Therefore, it is thought that negative reversion of CTC is difficult to achieve in patients with a large amount of bone metastasis at baseline, that is, in patients with a high BSI on bone scintigraphy (Table 2). In contrast, significant factors associated with cases showing negative reversion during ENZ administration were higher hemoglobin levels at baseline (Table 2). Based on the general concept, as cancer progresses, anemia worsens. This has been reported in many clinical studies regarding mCRPC with bone metastases [19,20]. Therefore, low hemoglobin levels are correlated with cancer progression and may be involved in the inability to achieve negative reversion of CTCs. Another significant factor associated with negative reversion was the absence of lymph node metastasis at baseline (0/12 cases) (Table 2). It remains unclear why the absence of lymph node metastasis was conducive to CTC disappearance from the blood with ENZ treatment. Lymph node metastases, in principle, spread through the lymphatics, and on the contrary, bone metastases are considered to progress hematogenously; however, CTC-positive cases with lymph node metastases at baseline are thought to progress through two pathways, hematogenous and lymphatic, which might contribute to treatment resistance. According to various studies examining the relationship between lymph node metastases and CTC counts in different cancer types, it has been reported that the number of CTCs at diagnosis is significantly higher in patients with metastatic lymph node cancer compared to those without metastatic lymph node cancer [21,22]. Hence, it is inferred that the two modes of cancer progression, lymphatic and hematogenous metastases, may not be independent but could synergistically promote progression. This mechanism suggests that the absence of lymph node metastasis might be a crucial condition for achieving negative reversion of CTCs in the blood.

The elimination of cancer cells from the blood during treatment had the strongest positive correlation with OS (Table 3). At baseline, higher hemoglobin levels were significantly correlated with prolonged OS (Table 3). This result is consistent with previous reports showing that pain, visceral metastasis, anemia, and bone scan progression are risk factors for poor prognosis in patients with mCRPC [23]. Among these, many reports have shown that anemia is a particularly important factor for poor prognosis of mCRPC patients [19,20]. It has been suggested that persistent detection of cancer cells in the blood despite treatment might be an even stronger factor for poor prognosis (Table 4).

CTC analysis has the potential for practical clinical application in CRPC treatment. The presence or absence of CTCs and the number of CTCs before treatment can predict treatment resistance [24]. Furthermore, patients who are ARV7-positive in CTC especially have a significantly shorter overall survival [25]. In the present study, the CTC-negative group had a significantly longer OS than the CTC-positive group did at baseline. Furthermore, Heller et al. reported that a decrease or negative reversion of CTCs three months after ENZ treatment was a stronger biomarker for predicting therapeutic effects and OS than a decrease in PSA [26]. The group analyzed and reported on five randomized phase 3 clinical trials. Koinis F et al. performed this same experiment before and after one cycle of cabazitaxel treatment, demonstrating similar results [27]. In contrast, an early decline in PSA levels at three months was important for predicting ENZ treatment effectiveness, as has been reported in the PREVAIL trial, in which −30% or more PSA decline within three months was associated with a greater likelihood of 5-year survival [28]. In this study, achieving a negative reversion of CTC during treatment for mCRPC gave a statistically significant prediction of OS prolongation (Table 3). In other words, the results of large-scale studies in Europe and the United States [26] were well reflected in the validation analysis of Asian subjects at our institution. The results from this study suggest that this method of chronological CTC analysis might more accurately predict the therapeutic effect of various treatments in patients with mCRPC than the previously widely used PSA and response to imaging.

In contrast to the relatively invasive nature of tissue biopsy, CTC analysis offers a less invasive alternative that allows for time-series examinations. Consequently, monitoring CTCs throughout the treatment process, as demonstrated in this study, could prove to be more clinically feasible in the future. It is important to take into account the count of CTCs when elucidating the association between AR-V7 status and prognosis [29]. The CellSearch method could detect CTCs in 69.5% of cases where the AdnaTest method yielded negative results for CTCs [30], implying that the sensitivity of the AdnaTest method might be lower compared to the CellSearch method. Nevertheless, in our study, we adopted the AdnaTest as a point-of-care testing (POCT) technique, emphasizing a straightforward and practical approach to conducting CTC analysis. Despite potentially lower sensitivity in CTC detection compared to the CellSearch method, the AdnaTest could still prove valuable in clinical settings as a POCT tool.

This study had several limitations. First, the observation period was short, and this was a small-scale, single-center clinical study; thus, the number of fatal events was small. Therefore, patients whose CTCs were negative at baseline had a median OS that did not reach fatal events within the observation period. Further studies with longer observation periods are warranted. Although we classified 60 patients based on the absence or presence of CTCs before ENZ administration and further analyzed the groups that showed negative reversion of CTCs during ENZ treatment, only 12 patients were included in this cohort. We attempted to verify the presence of surplus parameters in multivariate analysis and presented the results in Table 4. However, the insufficiency in event counts casts doubt on the reliability of the examination. The scarcity of events is believed to stem from an observation period that was not sufficiently long and a shortage of cases. While the bivariate COX proportional hazards analysis suggested the potential significance of CTC reversion as a factor contributing to OS extension (Table 4), further exploration through Cox proportional hazards modeling, analyzing the time-dependent nature of covariates, seemed promising for evaluating the adequacy of the Cox model assumptions and making adjustments as needed. However, the uniform timing set for evaluating CTC reversion at three months post-treatment in this study posed challenges for conducting this analysis successfully. Recognizing the potential presence of both early and delayed CTC-negative reversion groups, a future endeavor could involve multiple, sequential evaluations of CTC post-ENZ treatment to examine the correlation with survival rates, offering a more comprehensive understanding of the true clinical landscape. A larger sample size is required to support the conclusions of this study.

Second, this was a single-arm study that did not include a control group, and it is necessary to apply this method to other CRPC medicines for comparison. In addition, androgen receptor splice variant-7 (AR-V7)-positive cases have also been demonstrated to develop significant ENZ treatment resistance, which is consistent with previous reports [11]. It is also necessary to analyze and compare additional details of AR abnormalities, such as AR-V7, using liquid biopsies. Third, only the AdnaTest was used for CTC collection and analysis. If a highly sensitive method, such as CellSearch, was used to count CTCs, the number of CTC-negative groups would be considerably lower. The AdnaTest method may also have contributed to the higher survival rate in the CTC-negative group. Finally, although ENZ was the first-line treatment in most cases, the treatment lines were not standardized; thus, the analysis of OS could be biased by variations in the line of treatment.

## 5. Conclusions

Our study showed that mCRPC patients with bone metastases without CTC detection at baseline had a significantly longer OS than those with positive CTC at baseline. Moreover, patients who achieved CTC-negative reversion during treatment showed long-term OS. Thus, chronological CTC analysis might predict the therapeutic effect of various treatments in patients with mCRPC more accurately than the previously widely used PSA levels and imaging responses. The AdnaTest kit would be useful in clinical practice as a POCT tool for evaluating CTC status. Furthermore, the timely evaluation of CTC status might help physicians make a more reliable prognosis for mCRPC.

To further verify the conclusion, future studies may be designed with longer observation periods, larger sample sizes, more sensitive detection methods, and, if possible, a control group and standardized treatment.

## Figures and Tables

**Figure 1 cancers-16-00772-f001:**
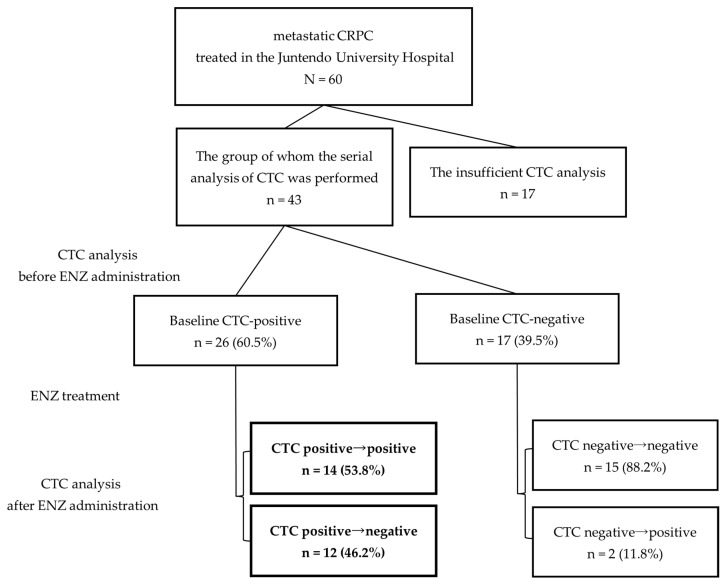
Flowchart of the analysis. CTC samples were collected from 46 patients at the time of ENZ administration and after starting ENZ treatment. The boxes indicate the number of patients who remained CTC positive, and those who experienced negative reversion are indicated in the box. CRPC, castration-resistant prostate cancer; CTC, circulating tumor cell; ENZ, enzalutamide.

**Figure 2 cancers-16-00772-f002:**
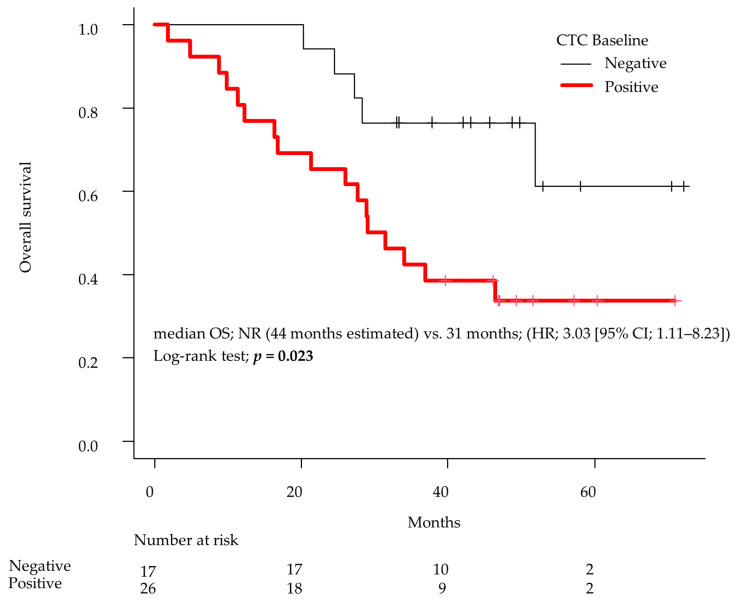
The Kaplan–Meier plots of overall survival. Overall survival (OS) was analyzed in the CTC-detectable and CTC-undetectable groups at baseline. Differences were compared using log-rank tests. CTCs, circulating tumor cells. The average observation period was 37 months (range 5–73 months).

**Figure 3 cancers-16-00772-f003:**
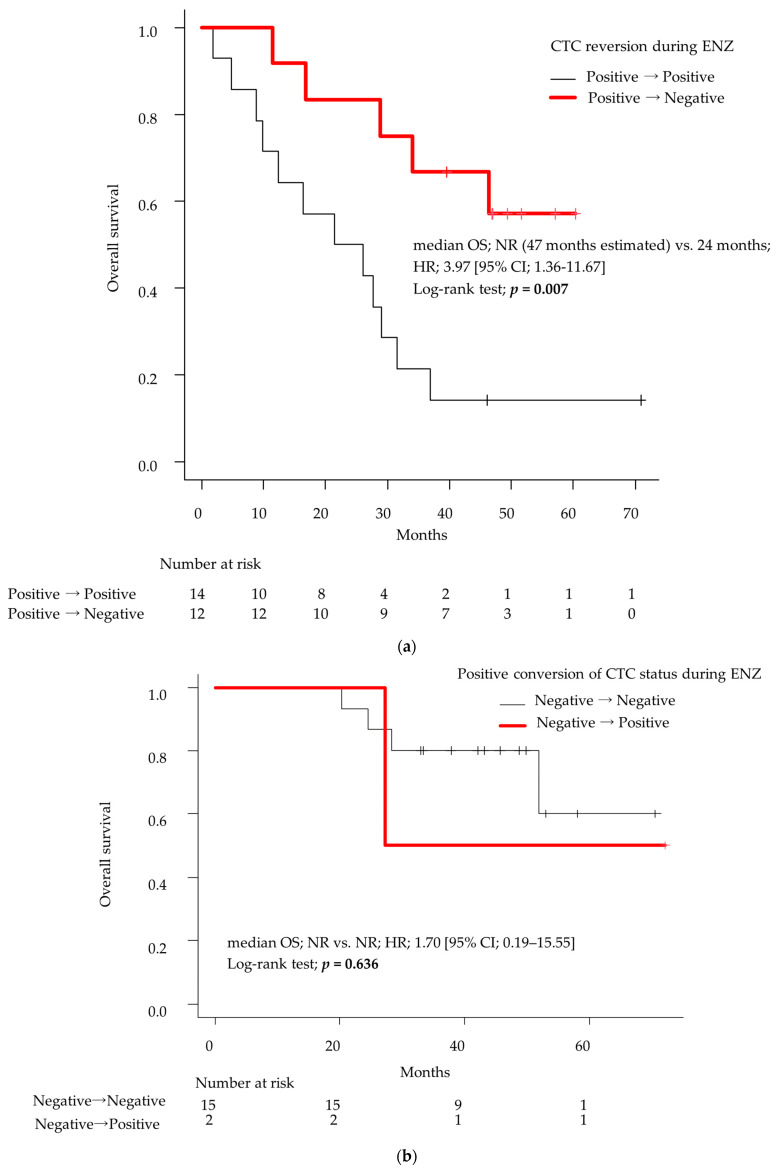
(**a**) Kaplan–Meier plots of overall survival of patients with baseline CTC-positive. The survival rate was compared between the following two groups: the group that achieved negative reversion of CTCs during ENZ treatment and the group that remained CTC-positive. The average observation period was 32 months (range 5–72 months). (**b**) The Kaplan–Meier plots of overall survival of patients with baseline CTC-negative. The survival rate was compared between the following two groups: the group that had positive conversion of CTCs during ENZ treatment and the group that remained CTC-negative. Differences were compared using log-rank tests. CTCs, circulating tumor cells; ENZ, enzalutamide.

**Table 1 cancers-16-00772-t001:** Patients’ baseline characteristics.

	All Patients	CTC Positive at Baseline	CTC Negative at Baseline	*p*-Value
Number of patients (n)	43	26	17	
Age median (IQR)	73.0 (69.0–68.0)	74.0 (70.3–77.8)	73.0 (68.0–78.0)	0.550
Gleason sum at diagnosis (6–7/8–10)	6–7:10 (23%)/8–10:33 (77%)	6–7:5 (19%)/8–10:21 (81%)	6–7:5 (29%)/8–10:12 (71%)	0.481
Body Mass Index median (IQR)	22.8 (21.0–24.5)	23.0 (21.0–25.0)	22.0 (21.0–23.0)	0.230
Family history of prostate cancer (y/n)	Yes 5 (12%)/No 19 (44%)/Unknown 19	Yes 4 (15%)/No 11 (42%)/Unknown 11	Yes 1 (1%)/No 8 (47%)/Unknown 6	0.615
Initial PSA median (IQR) (ng/mL)	119.2 (36.3–658.4)	335.2 (41.5–1981.6)	98.0 (18.3–332.4)	0.082
Baseline Bone Scan Index median (IQR) (%)	0.44 (0.11–1.31)	0.97 (0.36–1.65)	0.12 (0.10–0.29)	**0.008**
Presence of bone metastases at diagnosis (y/n)	Yes 37 (86%)/No 6 (14%)	Yes 22 (85%)/No 4 (15%)	Yes 15 (88%)/No 2 (12%)	>0.999
Presence of lung metastases (y/n)	Yes 4 (9%)/No 39 (91%)	Yes 2 (8%)/No 24 (92%)	Yes 2 (12%)/No 15 (88%)	>0.999
Presence of lymph node metastases (y/n)	Yes 10 (23%)/No 33 (77%)	Yes 6 (23%)/No 20 (77%)	Yes 4 (24%)/No 13 (76%)	>0.999
Radiation treatment (y/n)	Yes 6 (14%)/No 37 (86%)	Yes 2 (8%)/No 24 (92%)	Yes 4 (24%)/No 13 (76%)	0.193
Docetaxel before ENZ (y/n)	Yes 4 (9%)/No 39 (91%)	Yes 3 (12%)/No 23 (88%)	Yes 1 (6%)/No 16 (94%)	>0.999
Abiraterone before ENZ (y/n)	Yes 3 (7%)/No 40 (93%)	Yes 2 (8%)/No 24 (92%)	Yes 1 (6%)/No 16 (94%)	>0.999
Baseline alkaline phosphatase median (IQR) (U/L)	238.0 (158.5–354.0)	264.5 (178.5–387.5)	179.0 (148.0–243.0)	0.054
Baseline PSA median (IQR) (ng/mL)	9.6 (4.1–36.6)	26.2 (4.7–80.2)	4.4 (2.6–8.2)	**0.004**
Baseline serum calcium median (IQR) (mg/dL)	9.3 (9.0–9.5)	9.3 (9.0–9.7)	9.3 (9.0–9.4)	0.688
Baseline hemoglobin median (IQR) (mg/dL)	13.1 (11.6–13.7)	12.6 (11.3–13.6)	13.4 (12.7–13.8)	0.239
Time to CRPC median (IQR) (month)	16.0 (10.0–39.0)	15.5 (8.3–32.5)	16.0 (13.0–42.0)	0.258

The baseline characteristics of all patients before treatment change were indicated in the “All Patients” column. They were subsequently classified based on the absence or presence of CTCs at baseline, and the same variables were used to compare the CTC-positive and CTC-negative cohorts. Fisher’s exact test was used for categorical variables, and the Mann–Whitney U test was used for normally distributed continuous variables. Red bold indicates *p* < 0.05. IQR, interquartile range; CRPC, castration-resistant prostate cancer; CTCs, circulating tumor cells; ENZ, enzalutamide; PSA, prostate-specific antigen.

**Table 2 cancers-16-00772-t002:** A comparison of the clinical background of groups with CTC-negative reversion and those with CTC-positive continuation.

	CTC-Negative Reversion (N = 12)	CTC-Positive Continuation (N = 14)	*p*-Value
Number of patients (n)	12	14	
Age median (IQR)	72.5 (69.8–77.0)	74.0 (71.0–78.8)	0.328
Gleason sum at diagnosis (6–7/8–10)	6–7:2 (17%)/8–10:10 (83%)	6–7:3 (21%)/8–10:11 (79%)	>0.999
Body Mass Index median (IQR)	23.0 (21.5–27.2)	23.0 (21.0–23.8)	0.517
Family history of prostate cancer (y/n)	Yes 2 (17%)/No 7 (58%)/Unknown 4	Yes 2 (14%)/No 4 (29%)/Unknown 8	>0.999
Initial PSA median (IQR) (ng/mL)	471.5 (37.8–2765.1)	335.2 (50.5–1015.0)	0.837
Baseline Bone Scan Index, median (IQR) (%)	0.50 (0.1–1.1)	1.46 (0.7–2.7)	**0.021**
Presence of lung metastasis (y/n)	Yes 1 (8%)/No 11 (92%)	Yes 1 (7%)/No 13 (93%)	>0.999
Presence of lymph node metastases (y/n)	Yes 0 (0%)/No 12 (100%)	Yes 6 (43%)/No 8 (57%)	**0.017**
Radiation treatment (y/n)	Yes 1 (8%)/No 11 (92%)	Yes 0 (0%)/No 14 (100%)	>0.999
Docetaxel before ENZ(y/n)	Yes 1 (8%)/No 11 (92%)	Yes 2 (14%)/No 12 (86%)	>0.999
Abiraterone before ENZ (y/n)	Yes 2 (17%)/No 10 (83%)	Yes 0 (0%)/No 14 (100%)	0.483
Baseline alkaline phosphatase median (IQR) (U/L)	267.0 (188.8–359.8)	264.5 (178.5–628.3)	0.606
Baseline PSA median (IQR) (ng/mL)	18.1 (4.3–48.7)	35.1 (9.6–139.2)	0.258
Baseline serum calcium median (IQR) (mg/dL)	9.3 (9.1–9.7)	9.2 (8.8–9.4)	0.219
Baseline hemoglobin median (IQR) (mg/dL)	13.5 (12.2–14.4)	11.9 (11.0–12.9)	**0.047**
Time to CRPC median (IQR) (month)	19.0 (11.5–38.3)	10.5 (6.0–27.8)	0.303

Clinical background factors in the group with CTC-positive continuation and CTC-negative reversion during ENZ treatment. Fisher’s exact test was used for categorical variables, and the Mann–Whitney U test was used for normally distributed continuous variables. Red bold text indicates *p* < 0.05. IQR, interquartile range; CRPC, castration-resistant prostate cancer; CTC, circulation tumor cell; ENZ, enzalutamide; PSA, prostate-specific antigen.

**Table 3 cancers-16-00772-t003:** Predictive biomarkers of overall survival based on each clinical factor.

Variable	*p*-Value
Age (>74 y)	0.201
Family history of prostate cancer (y/n)	0.271
Initial PSA (>335.2 ng/mL)	0.165
Gleason sum at diagnosis (8–10/6–7)	0.306
Time to CRPC (<15.5 months)	0.702
Presence of bone metastasis at diagnosis (y/n)	0.478
Presence of lung metastasis (y/n)	0.749
Presence of lymph node metastases (y/n)	0.410
Use of denosumab (y/n)	0.286
Baseline PSA (>26.2 ng/mL)	0.532
Baseline Bone Scan Index (>0.97%)	0.560
Baseline alkaline phosphatase (>264.5 U/L)	0.235
Baseline serum calcium (>9.3 mg/dL)	0.268
Baseline hemoglobin (<12.6 mg/dL)	** **0.026** **
PSA at 3 months after ENZ (>5.1)	0.511
PSA 30%-decrease (y/n)	0.353
PSA 50%-decrease (y/n)	0.226
Bone Scan Index at 3 months after ENZ (>0.85%)	0.502
Alkaline phosphatase at 3 months after ENZ (>219.5 U/L)	0.405
Serum calcium at 3months after ENZ (>9.3mg/dL)	0.780
Hemoglobin at 3months after ENZ (<12.9 mg/dL)	0.085
**Positive continuation/CTC-negative reversion**	** **0.007** **

Each factor significantly contributing to the prolongation of overall survival was extracted using univariate analysis by log-rank test. The cut-off value for each factor was the median. Two biomarkers, higher hemoglobin at baseline and achieving negative reversion status of CTCs, were significantly associated with prolonged OS. Red bold text indicates *p* < 0.05. CTCs, circulating tumor cells; ENZ, enzalutamide; PSA, prostate-specific antigen.

**Table 4 cancers-16-00772-t004:** Bivariate COX proportional hazards analysis.

Variable	HR	95% CI	*p* Value
Positive continuation/CTC-negative reversion	3.51	1.14–10.8	0.029
Baseline hemoglobin; low/high (cut-off 12.6 mg/dL)	2.86	0.96–8.50	0.060

A bivariate COX proportional hazards analysis was performed on the two factors, low baseline hemoglobin and persistently positive CTC, which were significant factors for poor prognosis for OS in univariate analysis. CTCs, circulating tumor cells; HR, hazard ratio; CI, confidence interval.

## Data Availability

All data analyzed in this study can be provided by requesting data from the corresponding author, S.H.

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
