# Peer review of "The Detection and Negative Reversion of Circulating Tumor Cells as Prognostic Biomarkers for Metastatic Castration-Resistant Prostate Cancer with Bone Metastases Treated by Enzalutamide"

_cancers, 2024, doi:10.3390/cancers16040772_

Round 1

Reviewer 1 Report

Comments and Suggestions for Authors

  This study demonstrates clinical importance of CTC status and dynamics in patients with CRPC receiving enzalutamide. Not only pre-treatment biomarkers, but on-treatment biomarkers are valuable and useful for treatment decision.There are several issues to be considered to draw a conclusion.The draft contains many inappropriate points in results, figures, citations indicating that the data are unreliable. The authors need to pay more careful attention to your paper writing.

  1) This study compared the OS between CTC-positive and -negative at baseline, and between CTC-negative reversion and positive continuation. The univariate analysis was performed in the CTC-positive group at baseline (Table 3). The title is “Detection and negative reversion of …”. However, the authors focused on negative reversion of CTC as described as conclusion.  CTC-negative at baseline was also associated with favorable oncological outcome. The patients can be divided into four groups; 1) CTC-positive (baseline)>>negative (3 months after ENZ initiation), 2) CTC-positive>>-positive, 3) CTC negative >>negative, 4) CTC negative>>positive. To know the association between CTC status and prognosis, comparison of prognosis among 4 groups would be appropriate.

2) PSA kinetics, especially PSA reduction at 3 months after treatments, are useful prognostic marker (page 8, lines 244-) and PSA reduction more accurately predict prognosis than PSA value at one treatment point. PSA value at 3 months was included as a variable in Table 3. Why did the authors use 3-month PSA value instead of 3-month PSA reduction?

3)    The authors speculated that chronological CTC analysis might be more accurate predictor than PSA levels (page 10 lines 291-292). If the authors think so, the authors should show the data on the association between OS and 3-month PSA decline after enzalutamide initiation, which is easy to know. The results will prove that their speculation is correct or not.

 4)    The description about the association between lymph node metastasis and CTC positive continuation (page 9, line 222-) is hard to understand. For example, the following sentence” lymphatic cancer progression might be proportional to hematogenous cancer progression, which might lead to treatment resistance (lines 229-230)” is incomprehensible. No data suggested that CRPC cells in lymph node metastases were more resistant to enzalutamide, compared to those cells in other metastatic organs. Is there any difference in response of lymph node metastasis (change in size) to enzalutamide between cases with CTC negative reversion and those with positive continuation?

5)    Fig2 and Fig3: Median OS for CTC baseline-negative and CTC negative reversion group were described as 44 months and 47 months, respectively. However, these OS curves did not reach the median survival during follow-up periods.

 6)    The authors need to pay more careful attention to write a paper.

·         Page 9 lines 229-230 “lymphatic cancer progression might be proportional to hematogenous cancer progression, which might lead to treatment resistance [18, 19]”. I did not find above results in Ref19

·         In Ref14, that study focused on 14 CTC-positive patients at baseline and did not assess the association between PFS and the presence or absence of CTCs at baseline.

·         Ref23: This manuscript used baseline PSA value, not decrease in PSA (page 9, line 243).

·         Fig3: The numbers of the vertical and horizontal axis scale are incorrect. The numbers of “No at “ are also definitely wrong.

Author Response

Reviewer 1:

Comments and Suggestions for Authors

This study demonstrates clinical importance of CTC status and dynamics in patients with CRPC receiving enzalutamide. Not only pre-treatment biomarkers, but on-treatment biomarkers are valuable and useful for treatment decision. There are several issues to be considered to draw a conclusion. The draft contains many inappropriate points in results, figures, citations indicating that the data are unreliable. The authors need to pay more careful attention to your paper writing.

Reply:

Thank you for your evaluation. As you pointed out, there were minor errors in the description, such as misalignment of reference numbers. Once again, Thank you very much for your meticulous attention and valuable comments. We submitted the first draft for English proofreading, but we have now resubmitted it for proofreading and made some adjustments to the context.

This study compared the OS between CTC-positive and -negative at baseline, and between CTC-negative reversion and positive continuation. The univariate analysis was performed in the CTC-positive group at baseline (Table 3). The title is “Detection and negative reversion of …”. However, the authors focused on negative reversion of CTC as described as conclusion.  CTC-negative at baseline was also associated with favorable oncological outcome. The patients can be divided into four groups; 1) CTC-positive (baseline)>>negative (3 months after ENZ initiation) 2) CTC-positive>>-positive, 3) CTC negative >>negative, 4) CTC negative>>positive. To know the association between CTC status and prognosis, comparison of prognosis among 4 groups would be appropriate.

Reply:

This is an interesting perspective. Thank you for bringing it to our attention. Our study primarily focused on the baseline CTC-positive group, but as you correctly pointed out, the prognosis of the baseline CTC-negative group is also important. We had only two cases in which CTC status changed from negative to positive; one patient was alive during the observation period, and the other died of cancer (27 months later). The negative to negative group, as expected, had a good prognosis, with the median OS not yet reached. This data has been added to Figure 3b. The group that was CTC-negative at baseline and remained negative during the observation period also did not reach the median OS during the observation period. However, there was no statistically significant difference in survival between this group and the patients who achieved negative reversion from positive CTC to negative. We have also mentioned this in the text.

2) PSA kinetics, especially PSA reduction at 3 months after treatments, are useful prognostic marker (page 8, lines 244-) and PSA reduction more accurately predict prognosis than PSA value at one treatment point. PSA value at 3 months was included as a variable in Table 3. Why did the authors use 3-month PSA value instead of 3-month PSA reduction?

Reply:

Thank you for highlighting important analysis methods. We agree that the rate of decline is more crucial than the absolute value of PSA. We have added an analysis of cases where PSA reduction after 3 months achieved -30% and -50% to Table 3. Although the results showed a tendency for longer OS, there was no statistically significant difference. We have included this finding in the text.

3)    The authors speculated that chronological CTC analysis might be more accurate predictor than PSA levels (page 10 lines 291-292). If the authors think so, the authors should show the data on the association between OS and 3-month PSA decline after enzalutamide initiation, which is easy to know. The results will prove that their speculation is correct or not.

Reply:

In the study referenced below (Ref.[28]), the PSA decline rate after 3 months was a significant factor in prolonging OS in enzalutamide treatment for mCRPC. As you pointed out, a clear conclusion cannot be drawn unless this study also demonstrates the relationship between achieving PSA reduction after treatment and OS. Thank you for your advice. We have added an analysis of -30% and -50% PSA decline after 3 months and OS analysis to Table 3. The results indicated that a -50% PSA decline tended to be associated with longer OS, but the difference was not statistically significant.

[28] Armstrong, A.J. et al. Five-year survival prediction and safety outcomes with enzalutamide in men with chemotherapy-naïve metastatic castration-resistant prostate cancer from the PREVAIL Trial. Eur Urol. 2020, 78, 347–357.

 4)    The description about the association between lymph node metastasis and CTC positive continuation (page 9, line 222-) is hard to understand. For example, the following sentence” lymphatic cancer progression might be proportional to hematogenous cancer progression, which might lead to treatment resistance (lines 229-230)” is incomprehensible. No data suggested that CRPC cells in lymph node metastases were more resistant to enzalutamide, compared to those cells in other metastatic organs. Is there any difference in response of lymph node metastasis (change in size) to enzalutamide between cases with CTC negative reversion and those with positive continuation?

Reply:

Thank you for your insightful comments. In this study's results, all cases achieving negative reversion of CTCs were those without lymph node metastasis at the pretreatment baseline. Why the absence of lymph node metastasis is conducive to CTC disappearance from the blood with ENZ treatment remains a challenging interpretation. Consequently, our initial text was not clear to readers. We have revised it as follows:

“According to various studies examining the relationship between lymph node metastases and CTC counts in different cancer types, it has been reported that the number of CTCs at diagnosis is significantly higher in patients with metastatic lymph node cancer compared to those without metastatic lymph node cancer [21, 22]. Hence, it is inferred that the two modes of cancer progression, lymphatic and hematogenous metastases, may not be independent but could synergistically promote progression. This mechanism suggests that the absence of lymph node metastasis might be a crucial condition for achieving negative reversion of CTCs in the blood.”

Before treatment, lymph node metastases were mostly less than 15 mm in diameter and were not considered target lesions according to RECIST. Thus, even lesions showing a tendency to shrink were categorized as non-CR/Non-PD, complicating accurate response rate evaluation in imaging.

5)    Fig2 and Fig3: Median OS for CTC baseline-negative and CTC negative reversion group were described as 44 months and 47 months, respectively. However, these OS curves did not reach the median survival during follow-up periods.

Reply:

Thank you for your advice. Indeed, neither of the two groups reached the median OS, and the months listed were merely estimated values. This point was not sufficiently explained. We have revised the description to "NR (Not reached)" and added (estimated) regarding the number of months.

 6)    The authors need to pay more careful attention to write a paper.

Page 9 lines 229-230 “lymphatic cancer progression might be proportional to hematogenous cancer progression, which might lead to treatment resistance [18, 19]”. I did not find above results in Ref19

Reply:

Thank you for pointing this out. We apologize for the incorrect reference numbers, which were off by one from the actual ones, and for not noticing this earlier. We have adjusted the reference numbers and the order of references to be accurate. The correct references are now [21, 22] in the revised version.

  • In Ref14, that study focused on 14 CTC-positive patients at baseline and did not assess the association between PFS and the presence or absence of CTCs at baseline.

Reply:

The reference numbers were incorrect as mentioned above. We apologize for the simple mistake. The correct reference is [10] in the revised version.

  • Ref23: This manuscript used baseline PSA value, not decrease in PSA (page 9, line 243).

Reply:

As above, the reference numbers were incorrect. We didn't notice a simple mistake. The actually correct Ref. is [26] in the revised version.

  • Fig3: The numbers of the vertical and horizontal axis scale are incorrect. The numbers of “No at“ are also definitely wrong.

Reply:

We apologize for not noticing these details earlier. These were simple errors when converting the original PowerPoint data for submission to the Cancers, which we have now corrected.

Reviewer 2 Report

Comments and Suggestions for Authors

From a biostats and clinical epidemiology point of view, this manuscript has been well planned, executed and reported. Here are some suggestions for the Authors.

- line 92, have you collected data about cancer familiarity too? if yes, please add this risk factor to the general list of covariates

- line 124, rather than t-test, better to use a non-parametric approach like the Mann-Whitney test, here and all along the results

- median follow-up for the whole cohort is lacking

- the number of OS events is lacking

- at the light of the 2 last previous questions, you should carefully evaluate if it would be possible to estimate OS by the multivariate Cox PH regression model too, actually no similar approach has been proposed and even a bivariate model (Hb+CTC) could be of great help! 

- line 128, the stats software/release infos are lacking

Author Response

Reviewer 2:

Comments and Suggestions for Authors

From a biostats and clinical epidemiology point of view, this manuscript has been well planned, executed and reported. Here are some suggestions for the Authors.

Reply:

We appreciate your evaluation of our research. Below we list each of our revisions and responses.

- line 92, have you collected data about cancer familiarity too? if yes, please add this risk factor to the general list of covariates

Reply:

Thank you for your insightful suggestion. We have added an analysis of family history and found that of the 43 participants, only 5 had a clear family history of prostate cancer, and 19 had no family history. However, 19 cases had unknown details. Further analysis revealed that family history was not a significant factor in baseline CTC status or CTC negative reversion. We have included these data in Tables 1-3.

- line 124, rather than t-test, better to use a non-parametric approach like the Mann-Whitney test, here and all along the results

Reply:

Thank you for your advice on statistical analysis methods. Acknowledging the non-parametric nature of the data, we have switched to the Mann-Whitney test. Consequently, Tables 1 and 2 have been revised. The significant factors listed in Table 1 remain unchanged from the original version. In Table 2, we identified two new factors significantly correlated with CTC-negative reversion, namely high baseline hemoglobin and low Bone Scan Index (BSI) on bone scintigraphy. We have included these results and their interpretation in the text.

- median follow-up for the whole cohort is lacking

Reply:

The observation period extended from the initiation of ENZ treatment until death or the end of the study period, with an average duration of 37 months (ranging from 5 to 73 months). We have added this information to the text.

- the number of OS events is lacking

Reply:

We apologize for overlooking this detail in our initial submission. The omission of the number of events in Figure 3 was an error that occurred during the conversion of the original PowerPoint data for submission to the journal Cancers, and we have now corrected this.

- at the light of the 2 last previous questions, you should carefully evaluate if it would be possible to estimate OS by the multivariate Cox PH regression model too, actually no similar approach has been proposed and even a bivariate model (Hb+CTC) could be of great help! 

Reply:

Thank you for your insightful advice regarding further analysis. Our ultimate aim with this study is to estimate OS using a multivariate Cox Proportional Hazards (PH) regression model and to develop a nomogram for predicting OS, as outlined in the referenced study [28]. Unfortunately, the total number of participants in our study differed significantly from that in the PREVAIL trial, and the small number of participants and events in our study precluded meaningful analysis using this approach. We intend to address this in future research. Furthermore, we appreciate your suggestion regarding the bivariate model (Hb+CTC). We performed additional analyses using this model, and the results indicated that negative CTC conversion might contribute more to longer OS than high baseline hemoglobin levels. We have included these findings as Table 3b.

- line 128, the stats software/release infos are lacking

Reply:

Thank you for your comment. We have added a description of the EZR software used for statistical analyses, along with the relevant references [18].

Round 2

Reviewer 2 Report

Comments and Suggestions for Authors

- The number of OS events is still lacking (overall and stratified by CTC neg/pos), how many pts died by the last follow-up!? just for now, we can not check if the model is overparametrized or not

- what about time-dependence for Cox PH covariates? have you tested it?

- it would be interesting to check the median follow-up, stratified by CTC neg/pos too!

- line 133 typo,  biostatistics

Comments on the Quality of English Language

minor

Author Response

Comments and Suggestions for Authors

Thank you for reviewing the revised manuscript. We appreciate the insightful advice on the statistical analysis, and we are grateful for the guidance provided. In response to your suggestions, we have conducted additional statistical analyses using the existing data. However, due to the limited original sample size, the number of events, and the relatively short observation period, there were constraints in the analysis. We would like to address these points below.

- The number of OS events is still lacking (overall and stratified by CTC neg/pos), how many pts died by the last follow-up!? just for now, we can not check if the model is overparametrized or not

Reply:

Thank you for your valuable feedback. As mentioned in the initial draft's Discussion section, a significant challenge in this study stems from the limited overall case numbers and the brevity of the observation period, resulting in a scarcity of events. At the point of analysis conclusion, we verified the event counts concerning overall survival rates stratified by baseline circulating tumor cell (CTC) negativity/positivity. The outcomes revealed that the total death events were 22 out of 43 cases (51%), with 17 events out of 26 cases (65%) in the baseline CTC-positive group and 5 events out of 17 cases (29%) in the baseline CTC-negative group. This information has been incorporated into the manuscript.

We interpreted your observation about "the model is overparametrized or not" as indicating concern about the model having excess parameters for the data, making proper generalization challenging. Recognizing the importance of striking a balance with overfitting, we acknowledge the difficulty in validating whether the model possesses an appropriate level of complexity or if there are extraneous parameters, given the insufficiency in death event counts - 17 out of 26 cases in the baseline CTC-positive group and 5 out of 17 cases in the baseline CTC-negative group. We have incorporated this consideration into the Discussion section's limitations, as follows:

“We attempted to verify the presence of surplus parameters in multivariate analysis and presented the results in Table 3b. However, the insufficiency in event counts casts doubt on the reliability of the examination. The scarcity of events is believed to stem from an observation period that was not sufficiently long and a shortage of cases.”

- what about time-dependence for Cox PH covariates? have you tested it?

Reply:

Thank you for your valuable advice on the statistical analysis, which has been extremely insightful. Regarding the CTC-negative reversion group, we acknowledge the possibility of both early and delayed reversion, as well as variations in the observation period. To address this, we consider analyzing the time-dependent covariates in the Cox proportional hazards model. This would involve assessing the appropriateness of the Cox model assumptions and making necessary adjustments by modeling the time-dependent interaction term (covariate × time) to evaluate how the effect of the covariate changes over time. We have attempted this by modeling the interaction between covariates and time; however, the uniform timing set for the evaluation of CTC reversion at three months post-treatment posed challenges. Defining the "time for the time-dependent covariate to change from 0 to 1" as "the time from ENZ administration to CTC reversion" makes the analysis unfeasible. Furthermore, the limitations in adequately evaluating the time-dependent covariates in the Cox proportional hazards model in this study were attributed to insufficient follow-up duration, low event occurrence, and the uniform definition of criteria for judging efficacy. We have incorporated these considerations into the limitations paragraph of the Discussion section as follows:

"While the bivariate COX proportional hazards analysis suggested the potential significance of CTC reversion as a factor contributing to overall survival (OS) extension, further exploration through Cox proportional hazards modeling, analyzing the time-dependent nature of covariates, seemed promising for evaluating the adequacy of the Cox model assumptions and making adjustments as needed. However, the uniform timing set for evaluating CTC reversion at three months post-treatment in this study posed challenges for conducting this analysis successfully. Recognizing the potential presence of both early and delayed reversion groups, a future endeavor could involve multiple, sequential evaluations of CTC post-ENZ treatment to examine the correlation with survival rates, offering a more comprehensive understanding of the true clinical landscape."

- it would be interesting to check the median follow-up, stratified by CTC neg/pos too!

Reply:

Thank you for your guidance. As you advised, we find it intriguing to investigate the differences in the median follow-up durations for patients based on whether they are CTC-negative or CTC-positive at baseline. This approach provides valuable insights into the follow-up periods for patients with distinct CTC statuses, potentially deepening our understanding of this study. The final observation period, inclusive of instances of event occurrences, resulted in a median of 31 months (range: 5-72) for the baseline CTC-positive group and 44 months (range: 21-73) for the baseline CTC-negative group. Notably, the CTC-negative group exhibited a prolonged follow-up with observed event occurrences until the conclusion of the analysis. These details have been documented in the main text.

- line 133 typo, biostatistics

Reply:

Thank you for checking the details. We have corrected the spelling mistake.

Round 3

Reviewer 2 Report

Comments and Suggestions for Authors

Most concerns have been solved, mind the high rate of different mistakes all along the manuscript

Comments on the Quality of English Language

minor

Author Response

Comments and Suggestions for Authors

Most concerns have been solved, mind the high rate of different mistakes all along the manuscript.

Reply:

We sincerely appreciate your time and effort in revisiting our manuscript. We are thankful for the acceptance of most of our revisions. Feedback from both reviewers highlighted several inaccuracies in our initial draft, including errors in reference numbers and conversion bugs in the figures. We deeply regret any confusion this caused and extend our apologies to the reviewers. In this final revision, all authors have thoroughly reviewed the manuscript and data to ensure their accuracy. In this revised version, since we further have corrected several confusing expressions and typographical errors, we are confident that we have rectified all previous oversights. Moreover, we have addressed the points raised by the editorial team concerning the need for improvements in the English language expression. For your convenience, the revised sections have been highlighted in red. We kindly request a thorough review of these modifications. Thank you for your understanding and ongoing support throughout this revision process. We eagerly await your feedback and guidance.